# Flavonoids and the Risk of Gastric Cancer: An Exploratory Case-Control Study in the MCC-Spain Study

**DOI:** 10.3390/nu11050967

**Published:** 2019-04-27

**Authors:** Facundo Vitelli Storelli, Antonio José Molina, Raul Zamora-Ros, Tania Fernández-Villa, Vasiliki Roussou, Dora Romaguera, Nuria Aragonés, Mireia Obón-Santacana, Marcela Guevara, Inés Gómez-Acebo, Guillermo Fernández-Tardón, Ana Molina-Barceló, Rocío Olmedo-Requena, Rocío Capelo, María Dolores Chirlaque, Beatriz Pérez-Gómez, Victor Moreno, Jesús Castilla, María Rubín-García, Marina Pollán, Manolis Kogevinas, Juan Pablo Barrio Lera, Vicente Martín

**Affiliations:** 1Grupo de Investigación en Interacciones Gen-Ambiente y Salud (GIIGAS)/Instituto de Biomedicina (IBIOMED), Universidad de León, 24071 León, Spain; fvits@unileon.es (F.V.S.); ajmolt@unileon.es (A.J.M.); vasilirous@gmail.com (V.R.); mrubig02@estudiantes.unileon.es (M.R.-G.); jpbarl@unileon.es (J.P.B.L.); vicente.martin@unileon.es (V.M.); 2Unit of Nutrition and Cancer, Cancer Epidemiology Research Programme, Catalan Institute of Oncology (ICO), Bellvitge Biomedical Research Institute (IDIBELL), 08908 L’Hospitalet de Llobregat (Barcelona), Spain; raulzamoraros@gmail.com; 3Instituto de Investigación Sanitaria Illes Balears (IdISBa), Spain; dora.romaguera@isglobal.org; 4Instituto de Salud Global de Barcelona (ISGlobal), 08003 Barcelona, Spain; manolis.kogevinas@isglobal.org; 5CIBER Fisiopatología de la Obesidad y Nutrición (CIBEROBN), 28029 Madrid, Spain; 6Consortium for Biomedical Research in Epidemiology & Public Health (CIBER en Epidemiología y Salud Pública-CIBERESP), 28029 Madrid, Spain; nuria.aragones@salud.madrid.org (N.A.); mobon@iconcologia.net (M.O.-S.); mp.guevara.eslava@navarra.es (M.G.); ines.gomez@unican.es (I.G.A.); gftardon@uniovi.es (G.F.-T.); rocioolmedo@ugr.es (R.O.-R.); mdolores.chirlaque@carm.es (M.D.C.); bperez@isciii.es (B.P.-G.); v.moreno@iconcologia.net (V.M.); jcastilc@navarra.es (J.C.); mpollan@isciii.es (M.P.); 7Cancer Epidemiology Section, Public Health Division, Department of Health of Madrid, 28035 Madrid, Spain; 8Oncology Data Analytics Program (ODAP), Catalan Institute of Oncology (ICO), L’Hospitalet del Llobregat, 08003 Barcelona, Spain; 9ONCOBELL Program, Bellvitge Biomedical Research Institute (IDIBELL), L’Hospitalet de Llobregat, Spain; 10Public Health Institute of Navarra-IDISNA, 31003 Pamplona, Spain; 11University of Cantabria–IDIVAL, Santander, Spain; 12Cancer and Public Health Area, FISABIO-Public Health, 46020 Valencia, Spain; molina_anabar@gva.es; 13Department of Preventive Medicine and Public Health, University of Granada, 18071 Granada, Spain; 14Centro de Investigación en Recursos Naturales, Salud, y Medio Ambiente (RENSMA), Universidad de Huelva, 21071 Huelva, Spain; rocio.capelo@dbasp.uhu.es; 15Department of Epidemiology, Regional Health Council, IMIB-Arrixaca, Murcia University, 30007 Murcia, Spain; 16Department of Epidemiology of Chronic Diseases, National Centre for Epidemiology, Carlos III Institute of Health, 28029 Madrid, Spain; 17Oncology Data Analytics Program (ODAP), Catalan Institute of Oncology (ICO), L’Hospitalet del Llobregat, 08908 Barcelona, Spain; 18ONCOBELL Program, Bellvitge Biomedical Research Institute (IDIBELL), L’Hospitalet de Llobregat, 08908 Barcelona, Spain; 19Department of Clinical Sciences, Faculty of Medicine, University of Barcelona, 08907 Barcelona, Spain; 20IMIM (Hospital del Mar Medical Research Institute), 08003 Barcelona, Spain; 21Universitat Pompeu Fabra (UPF), Departament de Ciències Experimentals i de la Salut, 08002 Barcelona, Spain

**Keywords:** flavonoids, intake, gastric cancer, case-control study, MCC-Spain

## Abstract

Several epidemiological studies have investigated the association between the dietary flavonoid intake and gastric cancer (GC) risk; however, the results remain inconclusive. Investigating the relationship between the different classes of flavonoids and the histological types and origin of GC can be of interest to the research community. We used data from a population-based multi-case control study (MCC-Spain) obtained from 12 different regions of Spain. 2700 controls and 329 GC cases were included in this study. Odds ratios (ORs) were calculated using the mixed effects logistic regression considering quartiles of flavonoid intakes and log2. Flavonoid intake was associated with a lower GC risk (ORlog2 = 0.76; 95% CI = 0.65–0.89; ORq4vsq1 = 0.60; 95%CI = 0.40–0.89; ptrend = 0.007). Inverse and statistically significant associations were observed with anthocyanidins, chalcones, dihydroflavonols and flavan-3-ols. The isoflavanoid intake was positively associated with higher cancer risk, but without reaching a statistical significance. In general, no differences were observed in the GC risk according to the location and histological type. The flavonoid intake seems to be a protective factor against GC within the MCC-study. This effect may vary depending on the flavonoid class but not by the histological type and location of the tumor. Broader studies with larger sample size and greater geographical variability are necessary.

## 1. Introduction

Gastric cancer (GC) is one of the most relevant malignancies. It is the fifth most common cancer and the third in the mortality ranking world [1,2]. In many countries, a survival of five years after diagnosis is under 30% [3]. Incidence rates show a great geographical variability, probably caused by dietary patterns, genetic differences and exposure to carcinogens (e.g., *Helicobacter pylori* infection, smoking or processed meats) [1].

The intake of fruits and vegetables has been shown to be inversely related to cancer risk [4,5,6] and a high intake of fruits and vegetables is recommended for cancer prevention [7]. One of these beneficial effects has been partially attributed to the high flavonoids content of these foods [8]. In the case of GC, these evidences are limited.

Flavonoids are secondary metabolites present in a wide variety of plants, involved in photosynthesis, and in the defense against both pathogens and ultraviolet radiation. The flavonoids cannot be produced by the human body, so they must be ingested by eating different plant-based foods (such as fruits and vegetables). These compounds have been shown to present preventive properties for a wide variety of diseases, such as cancer, cardiovascular diseases [9] or osteoporosis [10]. Accordingly, flavonoids consumption could reduce the risk of cancer development through various mechanisms [11,12,13], such as protection against DNA damage [14], blocking specific carcinogen pathways [15], inducing apoptosis [16], modulating cellular proliferation, antioxidant properties, anti-inflammatory [17], inhibiting angiogenesis [18], and suppressing matrix metalloproteinases secretion and tumor invasive behavior [19]. In addition, some flavonoids have antimicrobial effects, inhibiting the growth of *H. pylori* [20,21].

Although several studies have evaluated the inverse relation between the flavonoid intake and the digestive tract cancer risk [8,22,23,24,25], the epidemiological evidence is still insufficient [8]. A recent meta-analysis concluded that dietary flavonoids could reduce the risk of GC in the European population but not in the US or Asia [23]. This difference between the studies in Asia, America and Europe, suggest that the heterogeneity between the results could be partially explained by the geographical variability of food consumption.

One of the possible causes of the lack of consistency in the results between populations could be the several flavonoid subclasses by GC subtype. There are differences among GC risk factors by the anatomical and histological subtype [26,27]. According to their chemical structure, flavonoids can be classified into six principal subclasses: Flavonols, flavones, flavanones, flavan-3-ols, anthocyanins and isoflavones [17,28].

We hypothesize that the intake of flavonoids is associated with different effects on gastric cancer according to the subclass of the flavonoid intake analyzed and the histological type or location of gastric cancer.

## 2. Materials and Methods

### 2.1. Study Population

The MCC-Spain is a large population-based multi-case control study evaluating environmental exposures and genetic factors of common tumors in Spain [29]. Clinical and epidemiological data from cases of histologically confirmed gastric tumors were collected from 2008 to 2013 in public hospitals. The subjects were previously informed about the study, and those who volunteered signed an informed consent for their participation. In the present study, we included 329 cases of GC and 2700 matched controls, Figure 1. The MCC-Spain study has been approved by the ethics committee of each area, and it also follows the Declaration of Helsinki and the Spanish Data Protection Act 1999 [30].

The inclusion criteria for cases were the following: Living for at least six months in the same area, and being 20–85 years old. Controls were selected randomly from lists of the primary care area from cases and they were matched by the age and area.

### 2.2. Variables and Data Collection

Cases and controls were interviewed by trained personnel, to collect information on sociodemographic factors, health behaviors (such as type, frequency and duration of physical activity, or smoking), medical conditions, previous treatments received, and personal and family history of cancer. In addition, a validated food frequency questionnaire (FFQ) [31] to collect dietary habits and past alcohol consumption (from 30 to 40 years old) was used. Data regarding the histological subtype (intestinal and diffuse) and anatomical location of the gastric tumors (cardia and non-cardia) were collected by qualified personnel.

### 2.3. Ethics Approval and Consent to Participate

The protocol of the MCC-Spain study was approved by the ethics committees of the participating centres. All participants were informed about the study objectives and signed an informed consent form. Confidentiality of the data was secured by removing personal identifiers in the datasets, (Ley orgánica 3/2018, de 5 de diciembre de Protección de datos personales y garantía de los derechos digitales). The database was registered in the Spanish Agency for Data Protection, number 2102672171. Permission to use the study database will be granted to researchers outside the study group, after revision and approval of each request by the Steering Committee. More details on the organization of the project can be found online at http://www.mccspain.org. More detailed information can be found in a previous article published by the MCC-Spain project research group [29]. The MCC-Spain study has been approved by the ethics committee of each area, and it also follows the Declaration of Helsinki.

### 2.4. Assessment of Flavonoid Intake

The validated MCC-Spain FFQ [31,32] included questions about the frequency of consumption of foods grouped under ten food categories with 140 item questionnaire: (1) Meat products (including items of poultry and eggs, pork, beef, lamb, fish and seafood, and precooked meat-derived food), (2) vegetables and legumes, (3) fruits and nuts, (4) dairy products, (5) cereals (including bread and pasta), (6) sauces and seasonings, (7) oils and fats, (8) sweets and snacks, (9) vitamins and mineral supplements, (10) alcoholic and other beverages.

The amount of each food consumed daily was estimated based on reference tables of servings of food. Similarly to other studies, if a given food was a mixture of several others (e.g., “gazpacho” or vegetable puree) the recipe was calculated, i.e., the contribution of each ingredient to the total content [33]. Total energy intake, as well as the consumption of the each food as mentioned above were also estimated. Moreover, some questions about general dietary habits were included in the questionnaire and were used to adjust the responses to the FFQ following the methodology described in Calvert et al. [34]

### 2.5. Analysis of Flavonoid Intake

The flavonoid subclasses considered in the present study (hereinafter referred to as ‘flavonoids’ if not otherwise stated) were anthocyanidins, chalcones, dihydrochalcones, dihydroflavonols, flavan-3-ols (also called flavanol monomers), flavanones, flavones, flavonols, isoflavonoids, and proanthocyanidins (also called flavanol oligomers and polymers).

A subset of 58 foods from the MCC-Spain FFQ were considered for analysis of the flavonoids intake, including vegetables and legumes, fruits, cereals, sweets and snacks, and alcoholic and other beverages. The dietary intake of flavonoids, expressed as aglycones, was estimated using the Phenol-Explorer database [35], except for proanthocyanidins, which were estimated using the most recent US Department of Agriculture (USDA) database on proanthocyanidins [36]. We have used aglycone equivalents for the flavonoid intake estimation, instead of considering the amount of all individual chemical flavonoid species (glucosides, glucuronides, etc.) reported in the food. The rationale behind this decision was to standardize data from the results of different analytical methods to facilitate cross-study comparisons [37].

Briefly, the process used to estimate the flavonoid contents involved looking for concordance among the food subset used and the foods included in the Phenol-Explorer database, and obtaining the aglycone equivalents for each flavonoid subclass and food. The flavonoid intake was calculated in milligrams per day, using the food consumption data from the FFQ and the aglycone flavonoid content of each food referred in the Phenol-Explorer database. No retention factors were applied in the calculation of the amount of flavonoids ingested. The data from Phenol-Explorer contain information on the flavonoid concentration obtained from both chromatography and chromatography after hydrolysis analytic methods.

Chromatography is the most used technique to estimate the concentration of polyphenols in foods, and their results are expressed as glycosides, which can be expressed in aglycones with a simple calculation. On the other hand, we find chromatography after hydrolysis, a less used method that expresses the result in aglycones [38].

The number of foods with flavonoid content obtained by the chromatography method is higher than chromatography after hydrolysis, but even so there are some foods or flavonoid classes of which we only have information by chromatography after hydrolysis. In order not lose information from them, we complete it with chromatography after hydrolysis when there is no information of chromatography. To avoid having the content of flavonoids expressed in two different units (glycoside and aglycone) we have transformed the glycoside values obtained from chromatography to aglycone, through the following equation (Ec.1). Given that the chromatography method does not provide information on the flavonoid content for all foods, this information was completed with data from chromatography after hydrolysis.
(1)$$P\;aglycone_{\frac{mg}{100mg}}=\left(\frac{\left[glucoside\right]\;}{Pm\;glucoside}\right)\times \;Pm\;aglycone\;\left(\mathrm{Ec}.1\right)\;$$

Flavonoid intakes were adjusted for the total energy intake [39] to obtain a measure concerning the isocaloric intake of flavonoids. The residues were added to the flavonoid consumption preset for an energy intake corresponding to the average intake between the control subjects stratified by gender.

Intakes concerning flavonoid quartiles were established according to the distribution among controls stratified by gender, and the lowest category of consumption was used as a reference.

### 2.6. Statistical Analysis

Descriptive statistics of the characteristics of the participants were obtained for controls and cases by both the tumor specific location and histology. The consumption of flavonoids were log_2_-transformed as data was right-skewed [40]. On the other hand, its interpretation is simpler, given that the odds ratio (OR) indicates the risk of GC when duplicating the intake. The flavonoid intake was analyzed both as categorical (grouping the intake of flavonoids into quartiles of their distribution among controls) and continuous (transformed to log_2_). 

ORs of GC and the corresponding 95% CI were adjusted using multivariate mixed logistic regression models to assess the association between the flavonoid intake and risk of cancer, including the study area as a random effect term. Other variables included in the model were age, socioeconomic status, alcohol consumption, smoking status, salt intake, body mass index (BMI), physical activity as metabolic equivalent task (MET), first-degree family history, red meat intake, vegetables intake, and total energy intake. The stata statistical software release 13 [41] was used for the mixed effects logistic regression, as well as the Python version 3.14 [42] and R version 3.6 [43] for the extraction of Phenol-Explorer web data on the polyphenol content in foods, and for the calculation of the flavonoid consumption by the individuals, respectively.

## 3. Results

The current study includes 329 GC cases and 2700 controls. The characteristics of cases and controls are shown in Table 1. The proportion of cases with the first-degree family history of GC (16.1%) was higher than controls (6.3%). Cases were more likely to have a higher intake of alcohol and salt. The daily amount of flavonoids ingested by all subjects is presented in Table 2, and no significant differences between controls and cases were found. We found a similar proportion of flavonoid-containing foods consumed in both controls and cases, with the lower overall intake of each flavonoid subclass in cases.

The results presented in Figure 2 are indicative of a statistically significant dose-dependent inverse association of flavonoids intake and GC risk (OR_log2_ = 0.76; 95%CI = 0.65–0.89. OR_q4vq1_ = 0.60; 95%CI = 0.40–0.89; p_trend_ = 0.007). A statistically significant inverse association was observed with the intake of anthocyanidins (OR_log2_ = 088; 95%CI = 0.80–0.96. OR_q4vq1_ = 0.68; 95%CI = 0.46–1.01; p_trend_ = 0.019); chalcones (OR_log2_= 0.89; 95%CI = 0.83–0.95. OR_q4vq1_ = 0.54; 95%CI = 0.36–0.81; p_trend_ = 0.001); dihydroflavonols (OR_log2_ = 0.89; 95%CI = 0.84–0.95. OR_q4vq1_ = 0.50; 95%CI = 0.34–0.75; p_trend_ = 0.001) and flavan-3-ols (OR_log2_ = 0.82; 95%CI = 0.73–0.92. OR_q4vq1_ = 0.56; 95%CI = 0.36–0.88; p_trend_ = 0.01). An inverse association was also observed with the intake of proanthocyanidins although the effect was only statistically significant when using the log_2_ transformation of the flavonoid intake (OR_log2_ = 0.82; 95%CI = 0.71–0.94) but not in the Q4 compared to Q1 analysis (OR_q4vq1_ = 0.91; 95%CI = 0.59–1.39; p_trend_ = 0.821). Dihydrochalcones intake was statistically significant positively associated with GC risk when was evaluated in quartiles, but not as a log_2_ (OR_log2_ = 1.02; 95%CI = 0.95–1.11. OR_q4vq1_ = 1.52; 95%CI = 1.02–2.28; p_trend_ = 0.02).

Adjusted ORs and 95% confidence intervals (CI) of GC according to the flavonoid intake. ORs were adjusted for age, gender, socioeconomic status, area of residence, GC family history, body mass index, smoking, physical activity, energy, sodium, red meat, vegetables and past alcohol intake.

No statistically significant differences have been observed in the behavior of the total intake of flavonoids and their respective subclasses for the different GC locations or histological subtypes (see Appendix A). Likewise, no statistically significant differences were observed in the behavior of the total intake of flavonoids and their respective subclasses for men and women separately but it has been observed that the differences are statistically significant only in men (see Appendix A).

## 4. Discussion

Several reviews have been published summarizing the epidemiological evidence on the inverse association between the dietary polyphenol intake and cancer risk, suggesting many biological benefits [24,44,45]. These benefits of flavonoids consumption could be related to lower GC risk through various mechanisms such as the antioxidant effect, blocking carcinogen pathways, inducing apoptosis, and inhibiting the growth of *H. pylori* [20,21]. Even low amounts of polyphenols can be continually absorbed and significantly increase the concentrations both at the plasma and cellular level [46]. Further, it has been observed that some classes of flavonoid can have an effect on cells even in µmolar concentrations and can interact on many molecular targets [47].

A case-control study conducted in Italy [25] mentions that the protective effect of fruit and vegetables on GC risk were attributable to a high consumption of flavonoids.

Our results suggest that the intake of total flavonoids could act as a protective factor in the development of GC, reducing the risk between 24% (log_2_) and 40% (Q4vsQ1). A meta-analysis on the flavonoid intake and GC has shown clear protective effects from the total intake of flavonoid [23] for the European population. Among the studies included, two of them found a significant protective effect of the flavonoid intake on all GC cases, regardless of histological type and location [22,48], by reducing the risk 19% (log_2_) in the Zamora-Ros’ study [22] and 51% (Q3vsQ1) in the Woo’ study [48].

In the analysis of the effect of the flavonoid intake by subclasses we have observed three different behaviors: Protective effect, null effect, and risk effect. Flavonoid subclasses with the protective effect include chalcones, dihydroflavonols, proanthocyanidins, anthocyanins, flavan-3-ols, flavanones.

This is the first study analyzing the effect of chalcones and dihydroflavonols on all cases of GC. A compound of chalcones, xanthumol, induced the cell death by the apoptosis and S phase cell cycle arrest [49,50], which could explain the protective effect of chalcones. But nowadays, there are no studies analyzing the effect of dihydroflavonols on GC in vivo, in vitro or in epidemiological trials. In the case of proanthocyanidins, we found a protective effect like in the Rossi et al. study [25]. An in vitro study indicated that proanthocyanidins are strong antioxidants with low cytotoxicity, which could play a relevant role in inducing the cancer cell apoptosis and cell cycle arrest [51].

Regarding the anthocyanins, a protective trend was reported in the study carried out by Woo et al. [48]. Among anthocyanins, delphinidin has been shown to have strong anticancer activities, possibly due to the suppression of the NF-κB pathway [52,53]. A study found that delphinidin significantly suppressed invasion and metastasis of lung cancer cells by downregulating the matrix metalloproteinase [54].

In the Woo et al. study, the flavonoid subclass flavan-3-ols showed a protective effect in all cases of GC, similar to the one described in our results. Flavan-3-ols have the most complex structures among subclasses of flavonoids. It has been found that several GC cell lines were sensitive to epicatechingallate (EGCG), a well-studied flavan-3-ol, by inducing apoptosis due to the inhibition of an anti-apoptotic protein [55]. Many signaling pathways might be affected by the EGCG treatment and it has been shown that EGCG exerts anti-proliferative effects in the GC cell by preventing the b-catenin oncogenic signaling pathway [56].

The results on flavanones obtained by Woo et al., agree with the results obtained in our study, indicating a protective effect of this subclass for all cases of GC. Concerningthe flavanones class, a statistically significant effect was found in the GC cells with naringenin, by inhibiting the cancer cell proliferation and migration, and inducing apoptosis, which might be related to its inhibition of the Akt signaling pathway [57]. Another study in colon cancer cells suggested that the pro-apoptotic activity of naringenin was mediated by the p38-dependent pathway [58]. Zhang et al., indicated that another treatment with flavanone, hesperetin, decreased cell proliferation and induced apoptosis through promoting the intracellular ROS accumulation [59].

On the other hand, our study shows that the flavonoid subclass called flavones seems to have no effect on GC. These results are not in line with other published articles such as Woo et al., whose study found a protective effect of this subclass; or Chen et al., whose study observed apoptotic effects of flavones in human GC cells in their in vitro. Referring to flavones, only one in vitro study has observed apoptotic effects in GC [60].

However, isoflavonoids and dihydrochalcones, were associated with an increased risk of GC. A study about the effect of the isoflavonoid intake and GC risk in Japan observed that this flavonoid subclass was not a significant risk factor of the GC risk in either men or woman [61]. On the contrary, in the Takayama study [62], their results show a non-significant effect of the isoflavonoid intake and GC risk in men, but showed statistically significant protective effects in women. The present study had a limited number of women diagnosed with GC, which did not allow us to draw conclusions. Furthermore, the differences in isoflavonoid intakes are outstanding: 1.4 mg/day in our study and 72.6 mg/day in the Japanese study. The results of the present study reveal an increased GC risk with increased dihydrochalcones intake, but no previous studies have evaluated this association. Therefore, results have to be interpreted with caution.

We did not find relevant differences in the effect on the different histological subtypes or localization of GC according to the intake of the different subclasses of flavonoids. Although the subtypes of GC have different risk factors [26], they share the probable protective effect of the intake of fruits and vegetables [27], which may explain our results. Nevertheless, the effect of flavonoids as a chemopreventive agent (that inhibits tumor promotion by inducing cell cycle arrest and by promoting apoptotic cell death) could protect all GC subtypes. The analyses carried out by the CUP [1] taking into account 24 studies on the consumption of fruits and vegetables in relation to the risk of developing GC indicate that a high consumption decreases the risk, without observing differences in the protection between anatomical subtypes. These results may be related to the protective effects of flavonoids, present in fruits and vegetables [1].

Contrary to our results, a Swiss study of the flavonol compound (quercetin) indicates a statistically significant protective effect in all GC and GC subtypes [63]. This difference in the results could be explained by the use of a different number of flavonols. In the present study we included all compounds of flavonols, whereas Ekstrom’s et al. study only included quercetin [63]. The EPIC study [22], reports the isoflavonoid intake similar to our data, suggesting a risk trend between the intestinal GC subtype for isoflavonoid. The lack of studies that analyze all cases, all flavonoid classes by location and histological subtype make the comparison difficult.

The evaluation of the effects of the flavonoids intake and cancer risk may be hindered by a number of underlying factors, including the difficulty to assess the bioavailability of flavonoids, differences in the estimation of their content in foods between database sources (USDA compared to Phenol-Explorer), the use of retention factors to take into account the cooking losses or the accuracy of FFQs. All these effects contribute to limit the accuracy of epidemiological studies. Furthermore, flavonoids are extensively metabolized within the human body after ingestion, both at the hepatic and colonic level, after interaction with the gut [64,65], which varies widely among individuals and could affect the response of the flavonoid consumption. We must take into account that a part of the variability between the results from different studies could be attributable to the heterogeneity of the dietary pattern of each country, since the flavonoid content in food can vary according to the different plants species, and environmental, geographic and storage conditions [64].

Regarding the tests of the effect of flavonoids in vivo and in vitro, these also have their limitations, and we only resort to these types of studies when we do not find epidemiological studies to discuss results. In vitro studies there are also numerous factors that are not contemplated, such as the effects of the microbiota on the “restructuring of polyphenols”, as well as their interaction on the ABC type transporters [66], the synergistic or antagonistic effects between the classes of flavonoids, since not only one class of flavonoid is consumed at the same time and, the interpersonal variation of the bioavailability of the flavonoids [67]. For these reasons, the results of in vivo, in vitro and epidemiological studies should be moderate in their conclusions since they all have their limitations, and explain the different results between studies.

The main sources of some flavonoids subclasses, such as chalcones, dihydrochalcones and dihydroflavonoids come from alcoholic beverages (wine and beer). According to some reports, reactive metabolites of alcohol are carcinogenic to humans [68]. This increased risk could be also explained by the production of prostaglandins, generation of oxygen radical species and lipid peroxidation. In addition, alcohol interferes with retinoid metabolism which could affect cellular growth, cellular differentiation and apoptosis. The WCRF/CUP found a statistically significant risk effect of 45 grams of alcohol consumed per day [1]. In our database, near 85% of individuals are below this value. Since the present study is based on a case-control study design, it is possible that the cases are applied as a lower consumption of alcohol than the real one, as a consequence these flavonoids appear as protective facts [69].

There are differences in the incidence of GC by gender, probably due to hormonal factors and life styles [26]. Even though no statistically significant differences were observed in the behavior of the total intake of flavonoids and their subclasses for men and women separately, we only found statistically significant results for men, probably due to the small number of female cancer cases included within this study.

Our study shares certain limitations known to other epidemiological studies related to the accuracy of the data inferred from indirect food intake reporting methods such as FFQs. Moreover, the aglycone flavonoids content in food was estimated without considering the retention factors, which could explain the losses during cooking. However, this information is not available for most of the individual flavonoids and further research is needed to fill the gaps within the databases.

On the contrary, this is the first study reporting the effect of dihydrochalcones and dihydroflavonols subclasses on GC stratifying by the anatomic and histological GC subtypes for men and women. The inclusion of the vegetable intake in the final-adjusted model could have also decreased the potential to find significant protective effects of flavonoids. Our models are over-adjusted by the energy intake, resulting in more restrictive outcomes, making it sometimes difficult to obtain significant associations between the flavonoid intake and GC. A complete database has been automatically generated including all available information on the flavonoid content in Phenol-Explorer, together with the mixing of data extracted from chromatography and chromatography after hydrolysis, the food ponderations of FFQ, and the high number of case and control records with all variables related with GC, resulting in an increased accuracy from the viewpoint of nutritional epidemiology. In addition, in this study, the information has been obtained in a similar way for cases and controls. We used a comprehensive methodology to evaluate the flavonoids intake, obtaining similar results to other case-control studies. Overall, our data on the total intake of flavonoid in the Spanish population sampled by the MCC-Spain study is consistent with previous reports [70].

Our results are in accordance with the recommendations made by CUP, which suggest that a low intake of fruit increases the risk of stomach cancer [1]. In this sense, one of the possible strategies to follow to reduce the risk of gastric cancer, and according to our results, is an increase in the consumption of fruits and vegetables rich in flavonoids.

## 5. Conclusions

As conclusion, results of this study suggest that the intake of flavonoids could influence the development of GC by lowering the risk, independently of the anatomic and histological subtypes of GC. We observed an inverse association between the flavonoid intake and risk of GC in men, but not in women, probably due to the insufficient sample size.

Further studies related to this research should consider a larger sample size, which could help identify more clearly the preventive role of the flavonoid intake on GC and elucidate the mechanisms underlying its etiology. 

## Figures and Tables

**Figure 1 nutrients-11-00967-f001:**
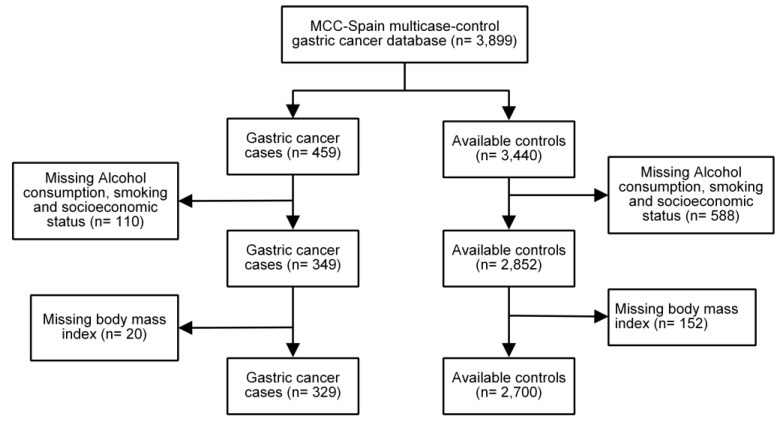
Algorithm for selection of controls and cases.

**Figure 2 nutrients-11-00967-f002:**
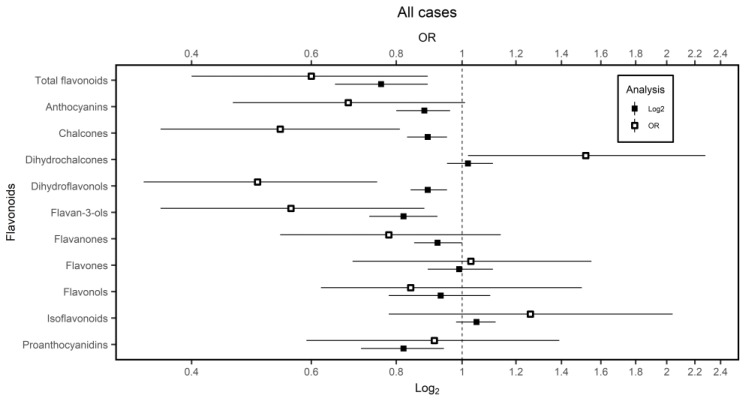
Gastric cancer (GC) risk according to flavonoids intakes in the multi-case control (MCC-Spain) study.

**Table 1 nutrients-11-00967-t001:** Characteristics of the study subjects.

Variables		Controls (%)	Cases (%)				
			Total	By Anatomical Subtypes		By Histological Subtypes	
		(*n* = 2 700)	(*n* = 329)	Cardia (*n* = 84)	Non-Cardia (*n* = 238)	Intestinal (*n* = 122)	Diffuse (*n* = 75)
**Age mean ± SD**		63.53 ± 0.21	65.40 ± 0.68	63.37 ± 1.26	66.06 ± 0.82	69.49 ± 0.96	61.79 (1.56)
**Gender (men, %)**		56.2	72.6	91.6	65.5	70.4	60.0
**Socioeconomic status**	High (%)	448 (16.59)	27 (8.21)	9 (10.71)	18 (7.56)	8 (6.56)	8 (10,67)
	Medium (%)	1 361 (50.41)	146 (44.38)	38 (45.24)	103 (43.28)	49 (40.16)	34 (45.33)
	Low (%)	891 (33.00)	156 (47.42)	37 (44.05)	117(49.16)	65 (53.28)	33 (44.00)
**Smoking status (%)**	yes	1 531 (56.70)	201 (61.09)	65 (77.38)	130 (54.62)	60 (49.18)	45 (60.00)
	no	1 169 (43.30)	128 (38.91)	19 (22.62)	108 (45.38)	62 (50.82)	30 (40.00)
**GC family history (%)**	yes	170 (6.30)	53 (16.11)	11 (13.0)	40 (16.81)	27 (22.13)	13 (17.33)
	no	2 530 (93.70)	276 (83.89)	73 (86.90)	198 (83.19)	95 (77.87)	62 (82.67)
**Physical activity (METS*h/week)**	<8	1 374 (50.89)	201 (61.09)	50 (59.52)	147 (61.76)	66 (54.10)	48 (64.00)
	≥8	1 326 (49.11)	128 (38.91)	34 (40.48)	91 (38.24)	56 (45.90)	27 (36.00)
**Body mass index (kg/m^2^)**	≤25	1 026 (38.00)	103 (31.31)	21 (25)	80 (33.61)	41 (33.61)	32 (42.67)
	>25–30	1 129 (41.81)	150 (45.59)	37 (44.05)	110 (46.22)	59 (48.36)	31 (41.33)
	≥30	545 (20.19)	76 (23.10)	26 (30.95)	48 (20.17)	22 (18.03)	12 (16.00)
**Alcohol consumption (g/day)**	0	418 (15.48)	47 (14.29)	8 (9.52)	39 (16.39)	22 (18.03)	13 (17.33)
	<12	1 179 (43.67)	103 (31.31)	16 (19.05)	84 (35.29)	40 (32.79)	27 (36.00)
	12–47	787 (29.15)	101 (30.70)	34 (40.48)	66 (27.73)	30 (24.59)	22 (29.33)
	>47	316 (11.70)	78 (23.71)	26 (30.95)	49 (20.59)	30 (24.59)	13 (17.33)
**Vegetables total intake (g/day), mean ± SD**		191.32 ± 2.42	180.80 ± 6.95	184.76 ± 18.32	177.88 ± 7.04	189.25 ± 9.7	185.63 ± 13.87
**Red meat intake (g/day), mean ± SD**		64.02 (0.76)	84.35 (2.94)	97.50 (6.54)	80.10 (3.28)	84.63 (4.58)	73.12 (5.19)
**Sodium intake (mg/day), mean ± SD**		3 008.64 (24.04)	3 529.30 (86.26)	3 758.56 (200.71)	3443.85 (94.85)	3 403.23 (144.63)	3 821.43 (187.65)
**Total flavonoid intake (mg/day), mean ± SD**		371.25 (3.72)	358.48 (9.88)	371.15 (18.63)	351.18 (11.69)	367.16 (14.72)	362.48 (24.51)

**Table 2 nutrients-11-00967-t002:** Average flavonoid consumption (mg/day) and percentage of total flavonoid subclass daily intake in all subjects of the study sorted by the three most consumed foods.

Flavonoids Subclass	Flavonoid Intake mg/day ± SD	Most Consumed Foods by Flavonoid Subclass Content-Gastric Cancer-All Controls					
		First (%)		Second (%)		Third (%)	
Anthocyanins	22.3 ± 0.35	Wine [Red]	31.3	Sweet cherry	24.7	Strawberry	15.9
Chalcones	0.004 ± 0.0001	Beer [Regular]	98.2	Beer [Alcohol free]	1.8	-	-
Dihydrochalcones	1.15 ± 0.023	Apple	95.3	Non-orange pure juice	5.5	-	-
Dihydroflavonols	1.87 ± 0.06	Wine [Red]	97.8	Wine [Rosé/White]	2.2	-	-
Flavan-3-ols	26.94 ± 0.33	Apple	33.4	Wine [Red]	20.55	Broad bean seed	13.64
Flavanones	43.81 ± 0.69	Orange/Tangerine	75.1	Orange pure juice	21.1	Tomato	1.48
Flavones	3.88 ± 0.08	Globe artichoke	24.7	Orange pure juice	18.0	Olives	16.16
Flavonols	23.75 ± 0.28	Swiss chard	34.2	Lentils/Chickpea/Common bean	15.9	Wine [Red]	10.7
Isoflavonoids	1.35 ± 0.11	Soy milk	95.3	Lentils	4.3	Nuts	0.27
Proanthocyanidins	200.99 ± 2.16	Beans	32.5	Apple	23.3	Wine [Red]	11.1

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
