# Peer review of "Flavonoids and the Risk of Gastric Cancer: An Exploratory Case-Control Study in the MCC-Spain Study"

_nutrients, 2019, doi:10.3390/nu11050967_

Round 1
Reviewer 1 Report
Minor
1. Please explain the hydrolysis method which the authors completed the information of flavonoid from chromatography.
2. Please write the full name of H. pylori in 66 line when it is described first time.
3. Please mention the reference in line 291(Sweden study), 294(Ekstrom’s et al study).
Author Response
Reviewer 1:
1-Please explain the hydrolysis method which the authors completed the information of flavonoid from chromatography.
We have modified the text to include the next explanation.
In the Phenol-Explorer database we can find information about the concentration of the different polyphenols classes, extracted by 2 different methods, chromatography and chromatography after hydrolysis, therefore quantifications have not been made by us. Chromatography is the most used technique to estimate the concentration of polyphenols in foods, and their results are expressed as glycosides, which can be expressed in aglycones with a simple calculation. On the other hand, we find chromatography after hydrolysis, a less used method that expresses the result in aglycones(1).
The number of foods with flavonoid content obtained by the chromatography method is higher than chromatography after hydrolisis, but even so there are some foods or flavonoid classes of which we only have information by chromatography after hydrolysis. To not lose information from them, when there is no information by chromatography, we complete it with chromatography after hydrolysis. To avoid having the content of flavonoids expressed in two different "units" (glycoside and aglycone) we have transformed the glycoside values obtained from chromatography to aglycone, through the following equation (ec.1).
2. Please write the full name of H. pylori in 66 line when it is described first time.
This was changed.
3. Please mention the reference in line 291(Sweden study), 294(Ekstrom’s et al study).
This was changed.
Reviewer 2 Report
The study titled "Flavonoids and the risk of gastric cancer: a case-control study in the MCC-Spain study" shows how higher intake of flavonoids may reduce the risk of gastric cancer. The manuscript has been written not good, the "Methods" need to be more clear with providing more details on the food frequency questionnaire, and the "Discussion" needs more work to appropriately discuss the results. Also, the manuscript needs to be extensively polished due to many grammatical errors and lack of coherence. Here are my comments:
While many in vitro and experimental studies have shown that different flavonoids can protect against molecular mechanisms involved in cancer and lower risk of different types of cancer, the authors have found that higher intake of isoflavonoids and dihydrochalcones was associated with an increased risk of GC. The rationale behind this should be discussed in the study.
I am really concerned about how the risk of gastric cancer could be associated with flavonoids intake after obtaining a food frequency questionnaire from research participants. Although the authors have mentioned some limitations of the study, it is not possible to control for confounding factors that may alter the risk of gastric cancer. Also, it is not clear in the manuscript how flavonoids intake of the subjects collected using a FFQ represent their usual intake over the course of 5-10 years. Cancer develops over time and many factors affect the initiation and progression of the disease and the association of the disease with specific foods such as flavonoids cannot be justified or judged by a cross-sectional study.
Author Response
Reviewer 2:
The manuscript has been written not good, the "Methods" need to be more clear with providing more details on the food frequency questionnaire, and the "Discussion" needs more work to appropriately discuss the results. Also, the manuscript needs to be extensively polished due to many grammatical errors and lack of coherence.
The English of the manuscript has been reviewed by a native person, so we hope you like it.
Two new citations have been included on the details of the FFQ and the discussion has been rewritten to fit the results even more.
Here are my comments:
While many in vitro and experimental studies have shown that different flavonoids can protect against molecular mechanisms involved in cancer and lower risk of different types of cancer, the authors have found that higher intake of isoflavonoids and dihydrochalcones was associated with an increased risk of GC. The rationale behind this should be discussed in the study.
We have modified the text to include the following explanation.
Since it is an epidemiological study, in the article we compare with other articles in the same field. Regarding the tests of the effect of flavonoids in vivo and in vitro, these also have their limitations, and we only resort to these types of studies when we do not find epidemiological studies to discuss results. In in vitrostudies there are also numerous factors that are not contemplated, such as the effects of the microbiota on the "restructuring of polyphenols", as well as their interaction on the ABC type transporters, the synergistic or antagonistic effects between the classes of flavonoids, since not only one class of flavonoid is consumed at the same time and, the interpersonal variation of the bioavailability of the flavonoids. At the same time, there are discrepancies in the results of the epidemiological, for example, with isoflavonoids, the lack of consumption of the main sources of this subclass is soybean whose consumption is not the same in European or Asian societies, being in this last much greater. For these reasons, the results of in vivo, in vitroand epidemiological studies should be moderate in their conclusions since all of them have their limitations.
I am really concerned about how the risk of gastric cancer could be associated with flavonoids intake after obtaining a food frequency questionnaire from research participants. Although the authors have mentioned some limitations of the study, it is not possible to control for confounding factors that may alter the risk of gastric cancer. Also, it is not clear in the manuscript how flavonoids intake of the subjects collected using a FFQ represent their usual intake over the course of 5-10 years. Cancer develops over time and many factors affect the initiation and progression of the disease and the association of the disease with specific foods such as flavonoids cannot be justified or judged by a cross-sectional study.
Your concern is very understandable. On the one hand, it is very complicated to take into account all the factors that may be involved in the development of gastric cancer, but as far as possible the epidemiological studies take these factors into account mathematically. This means that we generate mathematical models in which we include the variables that are known to be risk factors, such as alcohol consumption, physical activity, if they are smokers, etc. molecular variables such as mutations that occur within cells in the pathogenesis of cancer.
On the other hand, reference to the adjustment of the FFQ to reality over time, there are studies that indicate the stability of the diet over time to the cohort studies, to what despite the differences that exist in Diet changes are not significant (3).
References:
1 Garcia-Closas, R.; Garcia-Closas, M.; Kogevinas, M.; Malats, N.; Silverman, D.; Serra, C.; Tardon, A.; Carrato, A.; Castano-Vinyals, G.; Dosemeci, M., et al. Food, nutrient and heterocyclic amine intake and the risk of bladder cancer. European journal of cancer 2007, 43, 1731-1740, doi:10.1016/j.ejca.2007.05.007.
2 MCC-Spain questuionaries. Availabe online: (accessed on http://www.mccspain.org/wp-content/uploads/2016/07/Quest_MCCSpain.pdf.)
3 Goldbohm RA, van 't Veer P, van den Brandt PA, van 't Hof MA, Brants HA, Sturmans F, et al.Reproducibility of a food frequency questionnaire and stability of dietary habits determined from five annually repeated measurements. European journal of clinical nutrition 1995;49(6):420-9.
Reviewer 3 Report
I have now evaluated the article entitled: Flavonoids and the risk of gastric cancer: a case-control study in the MCC-Spain study. I found it very interesting and thanks the authors for their work and efforts. However, I was disappointed that they have not achieved successfully their objectives. I have the following points to be considered:
Their hypothesis "intake of flavonoids is associated with different effects on gastric cancer 93 according to the subclass of flavonoid intake analyzed and the histological type or location of gastric" seems to Null. Could the authors explain why?
How the results of this study compares to other world-wide studies to account for variations of food consumption?
From Table 2 and text, Chalcones intake was minimal ((0.004mg/d), yet the authors found it to be protective and reduce GC OR significantly (p=0.001)! Could they explain and comment on the concentration reaching the target tissues?
I would be interested if the authors can briefly compare the bioactivity of these various flavonoids with their bioavailability.
What about the combination effects of flavonoids on GC and cancer prevention in general. were the results obtained did include such effect? and can the authors add these rresults if they have them?
I tend to agree with the authors conclusion " results of this study suggest that the intake of flavonoids could influence the development of GC by lowering the risk of GC" However, there was no mention of mechanisms and strategy!
Minor points:
1. Group 1 Meat products should have been subdivided into White meat, Red meat (pork or lamb or beef), processed meat ...etc
2. Numbers in Figure 1 should be spaced or put a comma.
Author Response
Reviewer 3:
I have now evaluated the article entitled: Flavonoids and the risk of gastric cancer: a case-control study in the MCC-Spain study. I found it very interesting and thanks the authors for their work and efforts. However, I was disappointed that they have not achieved successfully their objectives. I have the following points to be considered:
Their hypothesis "intake of flavonoids is associated with different effects on gastric cancer 93 according to the subclass of flavonoid intake analyzed and the histological type or location of gastric" seems to Null. Could the authors explain why?
There are two aspects.
1º.- Type and location of the CG: it seems that the different histological types and localization of the CG may be due to different etiopathogenesis, so it is sensible to think that the effect of the flavonoids may be different according to location and histological type. The result has not been as expected, perhaps why flavonoids act in the same places of the pathogenesis of gastric cancer and do not depend on other factors.
2º.- The different types of flavonoids: I believe that they do not refer to this topic since we have verified different results.
How the results of this study compares to other world-wide studies to account for variations of food consumption?
Comparing studies from different parts of the world is complicated since, as you indicated, there are variations, for this purpose we try to compare studies as similar as possible, with similar flavonoid consumption, or studies with many subjects, for example the EPIC, or areas where get to have a similar diet. On the other hand, it is interesting to observe the heterogeneity in the effects of polyphenols according to the different geographical zones (1).
From Table 2 and text, Chalcones intake was minimal ((0.004mg/d), yet the authors found it to be protective and reduce GC OR significantly (p=0.001)! Could they explain and comment on the concentration reaching the target tissues?
Although its consumption is small, it has been observed that this class of flavonoid can have an effect on cells even in µmolar concentrations and can interact on many molecular targets (2). Since the consumption of chalcones in the models are as mg/d this value can have peaks of greater concentration, but maintaining a basal concentration similar to the average. Even low amounts of polyphenols can be “repeatedly” absorbed and can significantly increase the concentrations both at plasma and cellular level (3).
I would be interested if the authors can briefly compare the bioactivity of these various flavonoids with their bioavailability.
Although we currently have information on the bioactivity of flavonoids, information on their bioavailability remains scarce and only available for some foods (4). A review of the plasma concentrations of total metabolites ranged from 0 to 4 µmol/L, and the relative urinary excretion ranged from 0.3% to 43% of the ingested dose, depending on the polyphenol, but the data is incomplete and few study subjects in the bioavailability studies (5,6).
What about the combination effects of flavonoids on GC and cancer prevention in general. were the results obtained did include such effect? and can the authors add these rresults if they have them?
The effect of the combination of the flavonoids has been taken into account only with the consumption of the total flavonoids, but no effects of the combination of the flavonoid classes have been sought in this study, since we have focused mainly on the individual effects of each sub-case. This is partly because we are looking for new statistical methodologies that are more accurate to evaluate their effect since, when a food is consumed, not only a class of polyphenols is ingested, and this can give problems of collinearity of variables.
I tend to agree with the authors conclusion " results of this study suggest that the intake of flavonoids could influence the development of GC by lowering the risk of GC" However, there was no mention of mechanisms and strategy!
In the first place, we have only named specific mechanisms by which flavonoids can reduce the risk of GC because being an epidemiological study we can not discern through which one or all of the mechanisms that can interfere with flavonoids are those that are reducing the risk, since there are many beneficial effects of flavonoids. Regarding the strategy to follow to reduce the risk of gastric cancer, it would follow the line of the recommendations given by the World Cancer Research Fund, which indicates a greater consumption of vegetables and reduction in the consumption of red meat.
Minor points:
1. Group 1 Meat products should have been subdivided into White meat, Red meat (pork or lamb or beef), processed meat ...etc
This section has been better explained in the manuscript to clarify that the meat products group contains different separate items referring to red meats, white meats and others. In other words, not all meat products are treated as a whole.
2. Numbers in Figure 1 should be spaced or put a comma.
Thanks for your comment, we have changed it in Figure 1.
References:
1. Bo Y, Sun J, Wang M, Ding J, Lu Q, Yuan L. Dietary flavonoid intake and the risk of digestive tract cancers: a systematic review and meta-analysis. Scientific reports 2016;6:24836 doi 10.1038/srep24836.
2. Zhuang C, Zhang W, Sheng C, Zhang W, Xing C, Miao Z. Chalcone: A Privileged Structure in Medicinal Chemistry. Chemical reviews 2017;117(12):7762-810 doi 10.1021/acs.chemrev.7b00020.
3. Scalbert A, Williamson G. Dietary intake and bioavailability of polyphenols. The Journal of nutrition 2000;130(8S Suppl):2073S-85S doi 10.1093/jn/130.8.2073S.
4. Zamora-Ros R, Achaintre D, Rothwell JA, Rinaldi S, Assi N, Ferrari P, et al.Urinary excretions of 34 dietary polyphenols and their associations with lifestyle factors in the EPIC cohort study. Scientific reports 2016;6:26905 doi 10.1038/srep26905.
5. Manach C, Williamson G, Morand C, Scalbert A, Remesy C. Bioavailability and bioefficacy of polyphenols in humans. I. Review of 97 bioavailability studies. The American journal of clinical nutrition 2005;81(1 Suppl):230S-42S doi 10.1093/ajcn/81.1.230S.
6. Williamson G, Manach C. Bioavailability and bioefficacy of polyphenols in humans. II. Review of 93 intervention studies. The American journal of clinical nutrition 2005;81(1 Suppl):243S-55S doi 10.1093/ajcn/81.1.243S.
Round 2
Reviewer 2 Report
Thanks to the authors for responding to my questions and concerns.
Reviewer 3 Report
I have now the chance to look at the changes the authors have made in response to my suggestions, and I am satisfied that they have done their best to respond to my comments.
Therefore, I am happy to accept the manuscript in its present form provided that a smalll change made in the tittle. I suggest it too be:
Flavonoids and the risk of gastric cancer: an exploratory case-control study in the MCC-Spain study